# New Insights into the Role of PD-1 and Its Ligands in Allergic Disease

**DOI:** 10.3390/ijms222111898

**Published:** 2021-11-02

**Authors:** Miguel Angel Galván Morales, Josaphat Miguel Montero-Vargas, Juan Carlos Vizuet-de-Rueda, Luis M Teran

**Affiliations:** Department of Immunogenetics and Allergy, Instituto Nacional de Enfermedades Respiratorias Ismael Cosío Villegas, Calzada de Tlalpan 4502, Sección XVI, Tlalpan, Ciudad de México 14080, Mexico; jos.miguel.montero@gmail.com (J.M.M.-V.); janvizuet@hotmail.com (J.C.V.-d.-R.)

**Keywords:** allergy, PD-1, PD-L1, PD-L2, asthma, BTLA, rhinoconjunctivitis, atopic dermatitis, food allergy, anaphylaxis

## Abstract

Programmed cell death 1 (PD-1) and its ligands PD-L1 and PD-L2 are receptors that act in co-stimulatory and coinhibitory immune responses. Signaling the PD-1/PD-L1 or PD-L2 pathway is essential to regulate the inflammatory responses to infections, autoimmunity, and allergies, and it has been extensively studied in cancer. Allergic diseases include asthma, rhinoconjunctivitis, atopic dermatitis, drug allergy, and anaphylaxis. These overactive immune responses involve IgE-dependent activation and increased CD4+ T helper type 2 (Th2) lymphocytes. Recent studies have shown that PD-L1 and PD-L2 act to regulate T-cell activation and function. However, the main role of PD-1 and its ligands is to balance the immune response; however, the inflammatory process of allergic diseases is poorly understood. These immune checkpoint molecules can function as a brake or a kick-start to regulate the adaptive immune response. These findings suggest that PD-1 and its ligands may be a key factor in studying the exaggerated response in hypersensitivity reactions in allergies. This review summarizes the current understanding of the role of PD-1 and PD-L1 and PD-L2 pathway regulation in allergic diseases and how this immunomodulatory pathway is currently being targeted to develop novel therapeutic immunotherapy.

## 1. Introduction

Allergic diseases are characterized by hyperreactive immune responses to otherwise innocuous antigens in the general population, the incidence and prevalence of which are increasing worldwide, particularly in the developed world. In some western countries, every third child needs medical attention from a physician caused by allergic diseases [1,2,3].

The allergic inflammatory process involves different cell types that release a range of inflammatory mediators and cytokines. Th2 cytokines, including IL-4, IL-5, and IL-13, induce IgE production, while IL-5 promotes eosinophil infiltration [4,5]. Other factors, including Th1 cytokines, may sometimes affect the clinical manifestation creating a potential tolerance induction or hyperactivity [6]. Moreover, the progression of allergy disease is mediated by the innate immune system on the respiratory or gastrointestinal tract mucosa. Indeed, antigen-presenting cells (APCs) become activated, and epithelial alarmins (IL-25, IL-31, IL-33, and TSLP) are released due to activation of the airway epithelial cells in response to allergens [7], leading to activation of Th2/mast cell/eosinophil/eosinophil-mediated allergic pathology, including IL-4 and IL-5. Another pathway is the Th1/Th17/neutrophil-mediated response to chronic innate immune activation and irreversible airway obstruction [8,9]. In addition, the high cellular plasticity in the context of type-2-driven inflammation refers to the capacity of T-helper cell subpopulations to respond depending on environmental signals differentiating into other effector cell types. For example, in chronic asthma, Th1 and Th2 cytokines contribute to airway wall remodeling. Another group of cells that respond to environmental signals are group 2 innate lymphoid cells that exhibit a dynamic phenotype in allergic airway inflammation [10]. Allergic diseases such as asthma are heterogeneous with multiple conditions. The molecular phenotypes defined based on the predominant inflammatory cell are endotypes. In this context, the type 2 immune response can be categorized into different subgroups, such as IL-5-high, IL-13-high, or IgE-high, and others [11].

In contrast, IL-10 downregulates the inflammatory process. IL-10 acts directly on CD4+ T cells, primarily by down-regulating IL-2 and other cytokines produced by Th1 cells. It also inhibits Th2 cell production of IL-4 and IL-5. On dendritic cells (DCs), monocytes and macrophages, and APCs, the effects of IL-10 include inhibition of the production of free mediator molecules in the medium, such as receptor decreases, or abolished antigen presentation and phagocytosis [12]. Thus, IL-10 plays an essential role as a regulatory molecule in both innate and adaptive immune responses, leading to immune tolerance or dampening tissue inflammation in humans. Similarly, the programmed cell death 1 (PD-1) and its ligands PD-L1 and PD-L2 play an essential role in regulating T cells activation. It controls the immune balance preventing the accumulation of self-reactive T cells [13]. PD-1 has been characterized as a negative regulator of conventional CD4+ T cells. In addition, PD-L1 and PD-L2 have an essential but oppositive role in modulating and polarizing functions of T-cells in hyperreactivity [14]. This review provides an overview of the role of PD-1 and its ligands in allergic disease.

## 2. Molecular Mechanisms of Allergy

CD4+ T cells require three complementary signals to become fully activated—the T-cell receptor (TCR) signal, co-stimulation/inhibition signals, and cytokine priming. The first signal confers specificity to the immune response and plays an essential role in recognizing antigen presentation by MHC-II on the surface of APCs. However, this signal alone is not sufficient to fully activate these T cells. For full efficacy, T cells need a second, non-specific co-stimulatory signal. The coupling between APCs and naive T cells will signal surface molecules expressed on T cells and APCs. Both express co-stimulatory molecules such as CD80 and CD86, which belong to the B7 family. These molecular interactions modulate T cell function by binding to CD28 or cytotoxic T-lymphocyte-associated antigen 4 (CTLA-4) receptors; this interaction has been observed to have opposing effects depending on the different environmental signals. CTLA-4 inhibits T-cell responses and regulates peripheral T-cell tolerance. At the same time, CD28 promotes T-cell activation and survival. Finally, cytokines control differentiation into different effector cells that deliver signals such as IL-4, IL-5, IL-13, INFs, and proinflammatory cytokines [1]. This finding shows that these immunological checkpoint molecules have a role in fine-tuning the T-cell response by mediating stimulatory and inhibitory signals.

More recent studies have shown that numerous new co-stimulatory molecules have been described leading to the recognition that co-stimulation pathways are more complex than the classical two-signal model. Co-stimulatory molecules are the B7:CD28 family, such as CTLA-4 or programmed death (PD)-L1, and from the tumor necrosis factor receptor (TNFR) superfamily such as OX40 or CD27 (Figure 1) [15]. Likewise, immunological synapse interaction results in antigen recognition and T cell activation [16]. The TCR CD3 complex is expressed at the stable conjugation between a T cell and APCs. The immunological synapse consists of the binding of TCR/MHCII, CD2/CD58, CD4/CCR5 or CXCR4, CD8/TCR, and CD28, CD154, plus a central supramolecular complex (c-SMAC) activating CD80-86/CD28 or CTLA-4: CD80/CD86. The peripheral supramolecular complex (p-SMAC) Agrin and lysophosphatidic acid (LBPA), CD11a/CD18/leukocyte function-associated antigen-1 (LFA-1)/or intracellular adhesion molecule-1 (ICAM-1), F-actin and CD4/Lck and CD43/moesin, CD45 and F-actin [17,18]. All these checkpoint molecules affect T-cell activation and function of T cells and their survival. Research to better characterize the specific role of these molecules in allergy and asthma has not been the main focus of research in PD-1 and its ligands biology.

The triggering stage occurs in two phases, an early and a late phase. The early phase develops after a new exposure to the antigen and binds to antibodies attached to mast cells (MCs), basophils, or eosinophils. This phase causes the activation and releases rapidly of various preformed mediators and others synthesized de novo [19]. The mediators released by MCs and basophils lead to increased vascular permeability, vasodilatation, bronchial, and visceral smooth muscle contraction with local inflammation. The late phase develops without a new antigen presentation and occurs 72 h after the initial exposure. It involves the recruitment of cells, mainly eosinophils and Th2 lymphocytes, which increase mediators that maintain and exacerbate the inflammatory process [20].

Signaling through co-receptors is an essential mechanism for organizing and intimately subduing the immune response. The typical pattern of T-cell activation is carried out with positive signals given by human leukocyte antigens presented by HLA class I or II. Signals from co-receptors either enhance or prevent this activation. Other negative signaling molecules, such as CTLA-4, PD-1, and B- and T-lymphocyte attenuator (BTLA) (ICOS: H3, H4), are also involved. Moreover, inhibition of CD40-CD40L and CD6-CD166 receptors ceases to form other receptors such as CD49d/CD29-CD106 and numerous adhesion molecules and their ligands CD40-CD40L and CD6-CD166 CD11a/CD18-CD54 [14].

Th2 cells can be inactivated via PD-1 or with its ligands PD-L1 and PD-L2. PD-1 plays an important immunoregulatory role by reducing initial T cell activation, fine-tuning T cell differentiation, limiting self-reactive cells’ activation, and reducing the risk for autoimmunity and immunopathology [21]. The precise role of PD-1 and its ligands PD-L1 and PD-L2 is still being studied in allergic diseases.

## 3. PD-1 and Its Ligands PD-L1 and PD-L2

Research on PD-1 and its ligands has been performed mostly in cancer, and allergies have been under-investigated. Honjo and coworkers described PD-1 in 1992 [22]; a few years later, in 1999, its ligand PD-L1 was described by Lieping Chen [23]. PD1 is expressed on several cell types, including activated CD4+ and CD8+ T cells, B cells, natural killers (NKT), monocyte-macrophages, APCs, and tumor-infiltrating lymphocytes CD11c+ myeloid DCs [24,25,26]. PD-1 is primarily expressed on thymocytes during embryonic development and on CD4- and CD8- immature lymphocytes. It can also be expressed in naive lymphatic tissues (LTs), dormant LTs, eosinophils, basophils, and mast cells. It can also be expressed in other non-hematopoietic cell lines such as neurons, keratinocytes, endothelial cells, fibroblasts, placenta, retina, and hair follicles [27,28]. The common ɣ-chain cytokines IL-2, IL-7, IL-15, and IL-21, can induce PD-1 expression on T cells [29]. PD-1 has been related as a modulator of immune responses because PD-1–deficient mice developed autoimmune phenotypes [30]; also play a significant role in antigen presentation by DCs to CD8+ T cells by regulating ligand-induced TCR down-modulation [31].

Dong and colleagues described a third member of the B7 family, called B7-H1 (PD-L1). They reported that PD-L1 co-stimulates T cells via a receptor different from CD28, CTLA4, or ICOS (inducible co-stimulator) and delivers an activation signal to T cells [23]. PD-L1 protein is constitutively expressed on hematopoietic and non-hematopoietic cells such as DCs, granulocytes, T cells, B cells, and tumor cells [32,33,34]. In macrophages by LPS, IFN-γ, granulocyte-macrophage colony-stimulating factor (GM-CSF) and T cells, B cells plus DCs, the expression of PD-1 and its ligands are regulated by proinflammatory cytokines. IFNs, TNF-α. IL-2, IL-7 and IL-10, and IL-15 stimulate PD-L1 expression at the cell membrane. (Figure 2) [25]. A frequent administration of anti-PD-L1, such as atezolizumab, is required for cancer treatment because PD-L1 is high in circulating-myeloid cells [35]. The expression of PD-L1 within non-lymphoid tissues suggests that it may regulate the function of self-reactive immune cells in peripheral tissues or may regulate inflammatory responses [22]. PD-1 is an early brake that fine-tunes T-cell activation during antigen presentation after TCR signal transduction. PD-L1/PD-1 co-stimulation in antigen-presenting DCs contributes to PD-L1-induced TCR down-modulation [31]. Furthermore, PD-L1 stimulates the production of IL-10 by T-cells [23,36]. In contrast to PD-L1, the PD-L2 is rarely expressed on resting cells and can only be induced on DCs, macrophages, and bone marrow-derived cultured mast cells. The regulation of PD-L2 is by IL-4 and GM-CSF. PD-L2 can be induced on macrophages by IL-4 and IFNγ and DCs by anti-CD40, GM-CSF, IL-4, IFN-γ, and IL-12 [25,37].

PD-L1 and PD-L2 expression depend on distinct stimuli, and their expression patterns suggest both overlapping and differential roles in immune regulation [38,39]. These new data suggest opposing results reported regarding the PD-1 function [40]. Many cytokines play an essential role in inducing or maintaining PD-L1 expression. Both in vitro and in vivo studies have proved that the expression of PD-L1 protein in APCs, regulatory T lymphocytes, and cancer cells strongly relied on the existence of proinflammatory cytokines: IL-1β, TNF-α, IL-6, as well as IL-2, IL-7, IL-15, and IL-21. Furthermore, IFNs type I and type II are the second mechanisms driven by oncogenes (cyclin D-CDK4 cascade) [36,41,42,43,44]. The role of PD-L1 is to maintain peripheral tolerance and contribute to antigen presentation to T cells by dendritic cells [42]. PD-L1 is up-regulated in response to inflammation and suppresses excessive immune responses, which may cause unnecessary tissue injury. Tumor cells that arise from normal cells adopt this mechanism to evade tumor immunity. T-lymphocyte suppression for PD-L1 in the tumor microenvironment. Likewise, PD-L1 up-regulated to inhibit apoptosis in many cells and regulate glucose metabolism [45,46]. PD-1 and its ligands appear to play an essential role in maintaining T cell homeostasis in cancer and allergic diseases.

## 4. PD-1 and Its Ligands in Allergy Diseases

The type 2 immune response represents the main and the most clinically relevant immune response in allergic diseases. PD-1 and its ligands have an essential role in maintaining T cell homeostasis via regulatory T- B-cells (Tregs and Bregs) (Figure 3) [47]. The following section summarizes the new insights of PD-1 and its ligands, PD-L1 and PD-L2, in allergic diseases that include allergic asthma, skin immune response, rhinoconjunctivitis, food allergy, and anaphylaxis.

### 4.1. Allergic Rhinitis and Conjunctivitis

PD-1 and its ligand are involved in allergic airway disease. Nevertheless, little is known about the role of these immunomodulators in rhinoconjunctivitis. Both allergic rhinitis (AR) and allergic conjunctivitis (AC) are clinical conditions caused by inhalation or contact with an allergen, such as pollen, dust, mold, food, or epithelium of certain animals. Rhinoconjunctivitis has been recognized as a medical concern that affects up to 40% of the worldwide population and negatively impacts the patient’s quality of life [48,49,50]. Symptoms include inflammation of mucosa and sneezing, rhinorrhea, itchiness, nasal congestion, and are typically accompanied by inflammation of the conjunctiva and, in severe cases, the cornea [51]. Like other allergies, the pathogenesis of rhinoconjunctivitis is mediated by antigen-specific IgE antibodies produced by B cells that bind to the surface of mast cells and basophils. APC and CD4+ Th2 lymphocytes participate in sensitization to the allergen. Although, Th1, Th17, and Treg subsets cells also participate in the immune response (Figure 2) [52].

Analysis in murine models showed that antigen-specific CD4+ T cells play a critical role in developing AC by producing cytokines and inflammation [53,54]. In 2003, Fukushima et al. developed a mice model of experimental immune-mediated blepharoconjunctivitis (EC) to study CD4+ T-cell co-stimulatory, including PD-1 and its ligands. In this, PD-L1 and PDL-2 were blocked with specific antibodies (antagonistic anti-PD-1 mAb (RMP1-14, rat IgG2a), anti-PD-L1 mAb (MIH6, rat IgG2a), and anti-PD-L2 mAb (TY25, rat IgG2a) to evaluate the function after stimulation with pollen (short ragweed pollen (RW)). Pollen was injected intraperitoneally and rubbed into the eyes of the mice. They were then sacrificed to obtain serum, eyes, and spleen and detect eosinophil infiltrate, CD4+ T cells and DCs, as well as cytokines. Interestingly, treatment with anti-PD-1, anti-PD-L1, or anti-PD-L2 antibodies before challenge with RW significantly increased IL-4, IL-5, IL-10 and IL-13 by splenocytes, but not in the former conjunctiva. Therefore, treatment with anti-PD-1, anti-PD-L1, and anti-PD-L2 Ab affected immune system responses significantly. Experimental results concluded that PD-1 is a negative regulator of the development of conjunctivitis, which is mediated by CD4+ T cells. Treatment with anti-PD-1 did not affect allergic conjunctivitis during the induction or the effector phase with ragweed pollen, while anti-PD-L1 or PD-L2 affected the specific immune response during the induction phase. Nevertheless, it did not affect the eosinophil infiltration into the conjunctiva. Remarkably, anti-PD-L2 increased the ragweed-specific Th2 cytokine production and IgGs in serum, and eosinophil infiltration increased during the effector phase [55]. Further, an analysis of B and T lymphocytes attenuators (BTLA) found similarities with the PD-1 signal transducer, an antigen receptor tyrosine-based that inhibits T-cell activations by downregulating Th2 cytokine production. BTLA-deficient T cells show increased proliferation, the same as in PD-1^−/−^ mice [56,57].

On the other hand, AR is the most common allergic disease. It is an IgE-mediated inflammatory reaction mediated by the Th2 response. In an experiment with blood myeloid DCs and nasal tissue, biopsies from patients with AR displayed impaired expression of ICOS ligand/B7H2, and this defect contributes to the pro-allergic/Th2 activity. ICOS ligand belongs to the B7 family, co-stimulating molecules of T cell responses. The PD-1 and PD-L1 are elevated in DCs in mice with asthma and up-regulated T cells [58]. However, the expression of PD-L1 is not affected in AR patients. The blockade of PD-L1 resulted in up-regulation of IL-13 and IF-ɣ production. These results suggest that PD-L1 has a role in suppressive Th responses in AR patients [59].

Recently, Wang and Tan explored the association between the PD-1/PD-L1 pathway and Bregs from patients with AR [51]. Employing flow cytometry, the authors found that patients with AR have lower CD19+CD25+ Bregs than the healthy group. Similarly, the expression level of PD-L1 for this cell subset was decreased in AR patients showing a negative correlation with CD4+CXCR5 follicular T helper cells. Further, the levels of an immunosuppressive cytokine IL-10 were less expressed in AR patients. Additionally, anti-PD-L1 promoted Bregs apoptosis and inhibited the expression of IL-10 in CD19+CD25+ Bregs in vitro [51]. These results suggest that PD-L1 is a potential regulator in the treatment of AR. When it is less expressed, the immunosuppression function is weak, resulting in an increased immune response in AR.

A recent study compared the immune response to human rhinoviruses (HRV) infection between allergic and non-allergic patients with chronic rhinosinusitis using an air–liquid interface organ culture system to simulate HRV infection. The percentage of RSV infection was not significantly different between the allergic (74.4%) and non-allergic (72.7%) groups. However, allergy sufferers expressed mainly PD-L1 and IL-10 in the infected tissues, with a strong correlation between PD-L1 and IL-10 in the polarization. Conversely, non-allergic tissues showed increased IL-4 and IFN-γ [60]. Still, it is unclear how allergy factors work in the pathway underlying the PD-L1-mediated increases in IL-10 [61].

Evidence suggests that the expression of PD-1 and its ligands on the surface of immune cells in the nasal mucosa are higher in AR patients than in non-allergic patients. Interestingly, a positive correlation of the proportion of PD-1+CD4+ T cells with the visual analog scale (VAS) of AR and IgE serum concentrations was reported. The ratio of PD-1(+)CD4(+) T cells, PD-L1(+)myeloid DCs (mDCs), and Th2 cells in peripheral blood of AR patients were higher than the control group [62]. The PD-1/PD-L1 signaling pathway promotes the AR inflammatory response by inducing the Th2 type immune response. No differential expression was observed in PD-L2 in mDCs. Similarly, the expression of PD-1 and its ligands were investigated on tissue homogenates derived from patients with chronic rhinosinusitis and nasal polyposis (CRSwNP) compared with healthy controls using real-time quantitative PCR [63]. In this study, PD-1 mRNA expression was increased in nasal polyps of CRSwNP patients compared to controls. However, PD-L1/PD-L2 expression was significantly lower than in the control. These findings were confirmed by immunohistochemistry: PD-1^+^ cells were indeed increased in the submucosa of NP tissue, and it was localized to both T- and epithelial cells. In contrast, only a few PD-L1 and PD-L2 cells were expressed. Notably, PD-1 expression was associated with disease severity and tissue IL-5 expression suggesting an important role in the pathophysiology of CRSwNP. Interestingly, increased soluble PD-L1 (sPD-L1) has been found in the serum albumin AR patients, and levels of PD-L1 showed significant inverse correlations with eosinophil counts and disease severity, suggesting that PD-1/PD-L1 is differentially regulated in different compartments of the human body [64], leading to a downregulation of the allergic reaction.

### 4.2. Asthma

Asthma is a chronic disease of the airways involving various inflammatory cells and mediators [65]. It is characterized by bronchial hyperresponsiveness, chronic inflammation, and structural alterations in the airways [66]. The most common asthma symptoms are non-specific and include wheezing, shortness of breath, chest tightness, and cough. Asthma involves multiple phenotypes caused by a variety of mechanistic pathways (endotypes) and variable clinical presentations (phenotypes) [10]. Based on the level of expression of IL-4, IL-5, and IL-13 secreted by Th2-type cells such as CD4+ and lymphoid cells-type 2 (ILC-2), asthma endotypes are type 2 (T2) high or T2-low [67]. CD4+ cells are predominantly associated with allergic asthma and eosinophils, MCs, and basophils with airway inflammation. T cells play a critical role in the inflammation by secreting different cytokines following allergen exposure with an increment in IgE production [67,68]. Because of the inflammation of the airways, Th2 is considered crucial [69]. The asthma course is influenced by T cells subset and cytokines, such as NKT and Tregs. In contrast to Treg that has a protective role in asthma, Th17-associated cytokines can increase the severity [70,71,72].

PD-1 constitutive signaling can lead to apoptosis in vitro, but its primary function regulates physiological T cell activation. However, if PD-1/PD-L1 becomes chronic, it leads to CD4+ and CD8+ T cell anergy and exhaustion and tolerance [73,74]. Moreover, as mentioned above, it led to transient TCR down-modulation necessary for the exponential phase of T cell clonal expansion in physiological situations. It slows down the Ca(2+) flux and activation [31,75]. PD-L1 is a negative regulator of CD4+ and CD8+ T cells, which up-regulates PD-1 strongly after antigen presentation [31,76]. Meanwhile, in airway hyperreactivity (AHR), T cells are modulated and polarized in the opposite form by PD-L2. In the absence of PD-L2, asthma severity is increased. In contrast, PD-L1 deficiency reduces AHR [77].

Most studies of PD-1 in asthma have been performed in murine models. The blockade of PD-1/PD-L1 has distinct influences on different CD4+ T cell subsets that could explain some of the variable results observed on allergic asthma by the PD-1 family members [78]. These experiments illustrate that PD-1/PD-L1 blockade enhances AHR by developing a concomitant Th17 immune response [79]. On the other hand, lack of expression or blockade of PD-L2 resulted in increased AHR and lung inflammation, indicating that PD-L2 protects the lung against the initiation and progress of airway inflammation (Figure 3) [58,78].

Oflazoglu et al., in 2004, investigated the role of PD-L2 in mouse model asthma, which is the Th2-mediated immune response. They found that PD-L2 and PD-1 are up-regulated in the lung of mice challenged with ovalbumin. PD-L2 expression levels are high on DCs and PD-1 on T and B cells. Researchers also investigated the role of Th2 cytokines on asthma. Treatment in vitro of bone marrow-derived DC with some cytokines, as IFN-γ, IL-4, and IL-13, leading to increased expression of PD-L2. Recombinant PD-L2-Fc, which is fused with a mouse IgG Fc domain, was used to determine the function of PD-L2 in vivo and in vitro models. PD-L2-Fc in vitro only decreases the proliferation of pre-activated T cells and decreases IL-4, IL-5, and IL-13 levels. This experiment confirms the inhibitory role of PD-L2.

In contrast to in vitro results, in vivo administration of PD-L2-Fc showed an increase in T and B cell response, eosinophils, and lymphocytes infiltration airways, inducing an exacerbated inflammatory response. The authors comment that this difference between in vitro and in vivo results might reflect the importance of time interaction of PD-L2 with the cells, or simply a population of Tregs are targeted, allowing exacerbation in vivo. Another possibility is that recombinant PD-L2 can interact with a second receptor and enhance T cell activation [80].

PD-L2 deficiency increases the development of airway hyperresponsiveness and negatively regulates T cell activation. Meanwhile, PD-L1 seems crucial for developing airway hyperresponsiveness in a murine model [58,77]. In the literature, PD-L2 shows a suppressive effect that downregulates IL-4 and up-regulates IFN-γ, diminishing AHR. Contrary, PD-L1 has the opposite effect, up-regulating IL-4 and downregulating IFN-γ that favors Th2 inflammation and increases AHR [77]. A study in humans indicates that modulation of PD-1-mediated pathways is of little significance in PD-L1 expression or PD-L2 activation following allergen provocation in allergic asthma [81]. Bratke et al. analyzed how PD-1 and its ligands are expressed in an established human allergic asthma model of segmental allergen challenge (SAC). Classical serial methacholine and SAC inhalation challenge parameters were used, and skin reaction and serum IgE were measured. Peripheral blood lymphocytes and DCs (pDCs) and mDCs were obtained and identified by flow cytometry and ELISA detection of soluble PD-1, PD-L1, and PD-L2. The controls used belonged to the same patients but were taken before provocation. Then, pDCs (but not control pDCs) were loaded with human IgE and interleukin-3. CD4+ and CD8+ T cells exposed to SAC were significantly up-regulated. Moreover, PD-1 expression was significant in CD8+ T cells but not in CD4+ T cells that did not express PD-1, and controls did not express PD-1. In pDCs and mDCs challenged with SAC, PD-L1 was identified but PD-L2 only in mDCs that rapidly depleted. Soluble PD-1, PD-L1 and PD-L2 after SAC were not expressed; however, PD-L1 expression in pDCs after high-affinity IgE receptor cross-linking in bronchoalveolar lavage fluid (BALF) was significantly associated with PD-L1 expression in pDCs when IgE was inoculated. This demonstrated an association in low PD-1 expression by circulating CD4+ T cells and serum IgE concentrations in allergic asthma, with little change in the regulation of PD-L1 and PD-L2 [81].

Another significant aspect of PD-L1 is related to the generation of Tregs. It promotes the conversion of T cells into inducible T cells (iTregs), specifically from memory T cells [79,82]. T regs function as immunomodulators inhibiting Th2 cells, ICL2, and IgE-producing B cells in allergic asthma. Meanwhile, they produce tolerogenic DCs, regulatory B cells, and IgG4-producing B cells. The reduction of Tregs cells and the increase in Th2 cells initiates the development of AHR in allergic asthma [83,84]. Using a mice asthma model, McGee and colleagues have found that PD-1 has a role in reversing airway hyperresponsiveness and airway inflammation by iTregs [84].

In another mouse asthma model, Akbari and colleagues studied the role of PD-L2 on invariant natural killer cells (iNKT). Using double knockout mice PD-L2^−/−^, they studied cytokine production, lung inflammation, and AHR after the ovalbumin challenge. In the absence of PD-L2, AHR and lung inflammation increased and IL-4 increased by iNKT cells. In summary, PD-L2 downregulates iNKT function and cytokine production. In contrast, PD-L1^−/−^ mice showed a diminished AHR and lung inflammation, with an increase in IFN-γ production by iNKT cells [78]. These findings support the role of PD-L2 as an inhibitor and PD-L1 as a positive regulator of the immune response.

As explained earlier, ILC2s are an essential source of Th2 cytokines that promote AHR and lung inflammation. Helou and colleagues demonstrated that in airway inflammation induced by IL-33, the PD-1 limited the viability of ILC2s and downregulated their effector functions. Additionally, PD-1 can act as a metabolic checkpoint in ILC2s, affecting cellular activation and division. In a similar case, the employment of a PD-1 agonist diminished the ARH in a humanized mouse model of lung inflammation [85]. Indeed, the finding that a PD-1 agonist suppressed AHR and lung inflammation supports its potential clinical use in asthma treatment.

To date, there are no reports on the use of PD-1 agonists to treat asthma patients. However, elevated PD-1+ Treg and a significantly reduced PD-L1+ Treg were found in young asthmatics suffering from mildly persistent and moderate to severe asthma, suggesting an abnormality of the PD-1/PD-L1 signaling pathway resulted in the deficiency of Treg quantity. It is well understood that the PD-1/PD-L1 signaling pathway plays an important role in immune homeostasis through the promotion of Treg development and the inhibition of effector T (Th17). Thus, up-regulated PD-1 could interact with PD-L1 to suppress the activity of Treg cells, cytokine secretion, and cytotoxic capacity. On the other hand, compared with healthy controls, asthmatic children from the mildly persistent and moderately to severely persistent groups showed a significant increase in the percentage of PD-1+ Th17 and a decrease in the percentage of PDL1+ Th17, suggesting that the abnormality of the PD-1/ PD-L1 pathway was associated with the increase in Th17 cell quantity. The authors concluded that an altered PD-1/PD-L1 signaling pathway might favor an increase in the numbers of Th17 cells and lead to a decrease in Tregs, breaking the Treg/Th17 balance favoring the development of asthma. [86]. In a separate study, Moseyabian et al. reported an increased expression of PD-1 and Tim-3 (T cell Ig immunoglobulin and mucin domain containing-3) in CD4+ T cells of asthmatics [87]. Using the endobronchial allergen challenge (EAC) in asthmatic patients, Bratke et al. demonstrated a small decrease in endobronchial PD-1+ CD4+ T cells was accompanied by an increase in PD-L1 expression on endobronchial mDCs and pDCs, 24 h after AEC, which may favor Th2 inflammation [81]. These findings suggest that PD-L1 plays an essential role in immune checkpoint receptors and immune dysregulation of asthma.

PD-L2 level expression correlates with asthma severity in lung asthmatics biopsies. In mice, aeroallergen-challenged pulmonary mDCs have increased PD-L2 levels. When PD-L2 is blocked, AHR is diminished. This suggests that PD-L2 promotes AHR [88]. These results are similar to those of Oflazoglu, where PD-L2 seems to enhance AHR [80]. Nevertheless, these results differ from previous studies where PD-L2 inhibits AHR [78]. In summary, PD-L1 has a positive correlation with AHR, contrary to PD-L2 that has an inhibitory role. There is a contradictory result in the role of PD-1 and PD-L1/PD-L2. PD-L1 has distinct influences on different CD4+ T cell subsets, which may explain the variable results observed. Blockage of PD-L1 can be used to treat asthma. Meanwhile, the agonist of PD-1 and PD-L2 can be used to diminish the ARH.

### 4.3. Food Allergy

Atopic diseases usually occur in a progression called the atopic march. The main manifestations begin in infancy or early childhood and are present as atopic dermatitis, followed by the gradual development of food allergy, allergic rhinitis, and asthma. Food allergy is defined as an immune response to the ingestion of harmless food proteins. They are recognized because exposure to even minute amounts of these allergy-producing foods can trigger clinical conditions and gastrointestinal upset, urticaria, inflammation of the airways, and anaphylaxis, ranging in severity from mild to life-threatening [89]. The mucosal intestinal immune system typically exists in a state of active tolerance to food antigens and does not necessarily mount cellular or humoral immune responses to non-self-antigens due to regulatory mechanisms. One such mechanism is termed “oral tolerance”, referring to the natural development of induced tolerance to orally ingested stimuli in the gut-associated lymphoid tissue (GALT) [90]. Food proteins can have a tolerance induced in part by DCs residing in the intestinal mucosa and implemented by regulatory T cells, either CD4+CD25+Foxp3+ Tregs comprising thymus-derived autoreactive Tregs (nTregs), iTregs, and CD4+CD25-Foxp3- Tregs generated from specific naïve antigens. Data suggest that food allergies occur when the immune tolerance is disrupted and instead a sensitizing immune response characterized by plasmacytoid IgE production and initial presentation of food-specific antigen occurs. We have seen that experimental food allergy in mice, where tolerance testing has been performed, requires an adjuvant or the exploitation of alternative sensitization pathways to induce allergic sensitization [91].

In several tests and mouse models, Tregs play a central role in oral tolerance. One example is reported by Chinthrajah SR. et al., in 2016, in a model using the hapten 2,4-dinitrofluorobenzene (DNFB) where antibody depletion of CD25+ cells (a marker of Tregs) impaired oral tolerance commonly induced by DNFB feeding. However, transfer of CD4+ CD25+ cells (i.e., Treg cells) to CD4+ T cell-deficient mice, which do not typically develop oral tolerance following DNFB feeding, is sufficient to restore feeding-induced oral tolerance [92,93]. This means that PD-1 expression coupled with PD-L1 restores tolerance in mice. As mentioned above, mesenteric lymph node (MLN) DCs can induce Foxp3+ regulatory T cells to regulate immune responses to beneficial or non-harmful agents in the gut. One of these is tolerance to safe food and commensal bacteria. Several studies on DCs in MLNs have revealed that the CD103+ DC subset predominantly induces regulatory T cells, and PD-L1-deficient mice could not induce regulatory T cells in MLNs. One of the first studies was conducted by Matsumoto K. et al. and Oflazoglu E. et al. 2004, who generalized the data to refer to allergies as the two unique properties of PD-L2 suggested. First, as the suppression of Th2 responses is specific to PD-L2 but not to PD-L1, the disparate effects of PD-L2 blockade on allergies are not clear. Second, the disparate effects of PD-L1 and PD-L2 blockade on different immune responses and disease models raise the possibility that these ligands may preferentially regulate Th1 and Th2 responses. PD-L2 blockade is observed in the effector phase of the immune response. These findings suggest that the regulation of the PD-L2 activity may have therapeutic implications for controlling allergies in humans. Advances in the molecular and cellular biology of PD-L1 and PD-L2 should broaden our understanding of the mechanisms of allergic diseases [58,80].

Nevertheless, Fukaya et al. demonstrated in 2010 that MLN DCs should express PD-L1 and PD-L2 as a precondition for attenuating an antigen-specific CD4+ T cell response as a way of inducing de novo conversion of antigen-specific naive CD4-Foxp3 T cells into iTreg for active immune regulation and leading to the establishment of oral tolerance. They demonstrated that the antigen-specific immune response was increased in PD-L1^−/−^ and PD-L2^−/−^ mice or decreased in CD80^−/−^ CD86^−/−^ and B7-H2^−/−^ mice compared with WT mice after systemic immunization with an antigen. Therefore, they hypothesized that PD-L1 and PD-L2 might be essential for establishing oral tolerance [94].

Several mouse models were used to demonstrate the depletion of antigen-specific Foxp3+ cells, which can confer oral tolerance or induce regulatory T cells in MLN. Since Treg cells are very heterogeneous, the DEREG mice expressing the diphtheria toxin receptor under the control of the FoxP3+ promoter allows almost 90% depletion of FoxP3+ cells in various organs when injected with diphtheria toxin (DT). As a control, non-transgenic BALB/c mice were used with continuous oral ovalbumin (OVA) induction for tolerance. They also used OT-II transgenic mice that recognize the OVA model antigen in the context of MHCII and mice deficient in MADCAM1, a gene encoding for several binding proteins such as ITGB7 present in the intestinal venules and CX3CR1, the ligand required for the proliferation of TregFoxP3+ cells to the lamina propria (LP) present in local APCs. All these models were used to determine that MLN or intestinal tolerance to dietary antigens is determined in local lymph nodes and the conversion or migration of TregFoxP3+ cells. Nevertheless, the depletion of TregFoxP3+ cells in the lymph nodes generates the expansion or migration of peripheral TregFoxP3+ cells with the help of intestinal macrophages and the production of IL-10 [95]. Dietary antigens may promote differentiation of Tregs and the development of oral tolerance. Mice fed an elemental diet had reduced numbers of lamina propria Tregs, increased proliferation of Ag-specific T cells upon Ag feeding, and increased susceptibility to a model of allergic diarrhea compared to control mice fed standard chow [96].

The roles of individual Treg subtypes in oral tolerance are poorly explored, but Foxp3+ Tregs, such as Foxp3-, produce TGF-β and IL-10 and may be found in the gut, and many of the Foxp3- Tregs are probably type 1 regulatory T cells (Tr1) that migrate peripherally [92]. The roles of individual Treg subtypes in oral tolerance are poorly explored, but Foxp3+ Tregs, such as Foxp3-, produce TGF-β and IL-10 and may be found in the gut, and many of the Foxp3- Tregs are probably type 1 regulatory T cells (Tr1) that migrate from the peripherally [97]. These are some tests in mice that have preceded the description of the facts involving PD1/PD-L1. On the other hand, there is research on immuno-tolerance produced in cancer and autoimmunity trials where knockout mice have been used and for allergy treatments. There are also others in which inflammatory pathologies of the small and large intestine are described in which the behavior of immunosuppression is very similar to those of food allergies, as in the case of the reports by Bertolini et al. and Nakanishi et al. [90,98].

In already emerging major groups of allergic diseases, food allergies are an important group of allergic diseases. The rising situation in the pediatric age group is becoming a more significant concern. Some cases have been treated with food allergen avoidance or strict elimination diets and symptomatic treatment measures, which have not solved the problem. However, other severe symptomatology, such as anaphylaxis, can be life-threatening. Furthermore, the undesirable results of life-limiting diets can result in poor digestive disorders and malnutrition. One of the recent essential aspects of the study of allergy mediators has emerged from allergen-specific immunotherapy (AIT), which is the primary treatment for curing allergic disorders. Food allergy is a new field for AIT with promising results [99,100]. However, it has also elucidated the role of several mediators [19]. During IAT, IL-10, TGF-b, CTLA-4, and PD-1 are induced within the mycomedium. For example, there are reports of children hypersensitive to nuts, in this case, peanuts. Fifty-seven children randomized to the active oral immunotherapy dose (OIT) of peanuts were studied and were primarily sensitized to peanuts with a geometric mean (min, max) sIgE at Ara h 2 of 56.2 (0.82, 492.0) kUA/L and had the lowest observed mild adverse effect (LOAEL) of 18.4 (11.8, 28.6) mg peanut protein, and 78.9% had a history of peanut anaphylaxis. Control groups of patients with placebo doses were also used [101]. The trial was named TAKE-AWAY. This study with high dose (MMD) and minimum dose (IMD) peanut sIgG4/sIgE ratio was the only significant predictor for achieving MMD and suppression in the presence of anaphylaxis. For IMD, low doses of sIgG4/sIgE. In total, 75.5% of children with peanut anaphylaxis achieved a maintenance dose of 0.25–5 g, which concludes the success of the therapy. These findings are cited by Akdis and colleagues [19,102]. They conclude that IL-10, TGF-b, CTLA-4, and PD-1 are present to induce tolerance. We suspect that although these data are claimed to be sufficient to see tolerance, the role of PD-1 in these cases is unclear.

### 4.4. Skin Immune Response

The role in the skin immune response of the PD-1/PD-L1 pathway has not been thoroughly studied so far. There are few reports on the role of PD-1 and its ligand in macular reactions, erythematous papules, rash, and desquamation. Skin immune reactions are associated with an allergic response to medication, food, insect bites, and others. Typical skin allergic conditions include atopic dermatitis (eczema) and urticaria (hives); although, the hypersensitive response can trigger even an anaphylactic reaction with medical concerns leading to death [103]. More recently, there has been increased interest in adverse skin reactions in PD-1/PD-L1 checkpoint inhibitors (CPIs) [104,105,106].

Several studies have evaluated the effect of PD-1/PD-L1 on skin immune response using murine models. Recently, Bonamichi-Santos et al., 2021 analyzed the effect of PD-L1 blockade in a murine model of active cutaneous anaphylaxis (ACA). Mice treated with anti-PD-L1 during the sensitization phase showed a reduction in IgE and IgG1 levels. Furthermore, the allergic reaction intensity and MCs degranulation in the tissue were reduced but not for the challenge phase [103]. Tanaka and colleagues also investigated the contribution of PD-1/PD-L1 and PD-L2 in the regulation of the Th1-, Th2-, and Th17-type immune response using isolated APCs and three murine models with different types of inflammatory dermatitis. In vitro experiments showed that APCs stimulated by IFN-ɣ o IL-17A strongly up-regulates PD-L1 expression. On the contrary, PD-L2 was only up-regulated with IL-4. PD-L1-deficient mice had more severe changes in ear thickness for Th1- and Th17A-type immunity models. For the Th2-type model, a more severe change was observed for PD-L2-deficient mice [107]. Previous reports showed that when blocking PD-1 and PD-L1 with antibody, sensitivity was enhanced, and ear swelling in mice displayed hapten-induced contact hypersensitivity [107]. These results suggest that PD-1 and PD-L1 interaction play an essential role in attenuating Th1- and Th17-type immune responses.

Blocking of PD-1/PD-L1 with monoclonal antibodies has been used for different cancer treatments. Examples of these checkpoints inhibitors (CPIs) include nivolumab, pembrolizumab, cemiplimab, atezolizumab, avelumab, and durvalumab [108]. Although CPIs present good results, the patients frequently develop cutaneous immune-related adverse events (irAEs). T-cell activation is related to inflammatory dermatologic reactions that appear a few weeks after treatment. Typical manifestations include maculopapular rash, pruritus, psoriasis, lichenoid eruptions, bullous pemphigoid eruptions, hypopigmentation, and erythema [104,109,110]. Skin irAEs in patients treated with anti-PD-1/PD-L1 are mild to moderate with relatively low incidence (25%). Nevertheless, the combination with other inhibitors such as CTLA-4 showed an increased incidence of irAEs (>40%) [110,111].

Most cutaneous irAEs are related to Th1-type immunity. In addition, few patients develop primitive Th2-type immunity-related allergic diseases, such as urticaria, asthma, and AR as irAEs when treated with agents blocking the PD-1/PD-L1 axis. However, the exact pathogenesis of the immune response is not clear. In maculopapular rash, infiltration of CD4+ T cells with eosinophils occurs. Meanwhile, for lichenoid dermatitis, an increased abundance of CD163+ cells were observed [112].

In comparison, bullous pemphigoid eruption is characterized by deposits of immunoglobulin G and complement component C3 [113]. Pruritus occurs in approximately 20% of patients treated with anti-PD-1/PD-L1 therapy, associated with a skin rash. Melanoma patients treated with anti-PD-1 have shown hypopigmentation/depigmentation (vitiligo-like). The antitumor response is mediated by CD8+ cytotoxic T cells, the activation against melanoma-associated antigens shared by melanocytes and melanomas [114]. In contrast, in psoriasis, augmented helper T cells, such as Th1 and Th17, play a pivotal role in this pathogenesis [115,116]. Other severe skin clinical manifestations are mediated by the expression of PD-L1 in keratinocytes, leading to the targeting of apoptosis of cells by activated cytotoxic CD8+ T cells. This phenotype is accompanied by the upregulation of inflammatory chemokines [105,117].

### 4.5. Anaphylaxis

Anaphylaxis is a severe acute life-threatening multi-system reaction resulting from releasing many mediators from MCs, culminating in respiratory severe, cardiovascular, and mucocutaneous manifestations that can be fatal. Anaphylaxis is a severe immediate systemic hypersensitivity allergic reaction promoted by CM with antigen encounters and prompt degranulation of basophils [118]. The most recent sources show that drugs and food cause the incidence and mortality of anaphylactic reactions. Several animal models have been developed to study induced cutaneous anaphylaxis. Anaphylaxis has been reproduced and analyzed in murine models due to the practicality of breeding, reproduction, maintenance, and handling of these animals and their availability, including knockout and transgenic models [119]. Mouse models offer many advantages for studying food allergies, mainly that sensitization or allergic tolerance to specific allergens can be induced under controlled conditions, which is not possible in humans. Some immune mechanisms responsible for the breakdown of oral tolerance are not fully understood, but recent research suggests that alterations in Treg cell function and environmental factors, such as the microbiota, contribute to food allergic sensitization. Evidence in mice suggests that either the respiratory tract or via the skin successfully triggered allergic sensitization and anaphylaxis to several food antigens, including egg, peanuts, and hazelnuts [120]. Mice exposed cutaneously to hazelnut protein exhibited systemic Treg cell presence with probable PD-L2 expression concomitant with sustained IL-4 secretion over several months, implying persistent clinical sensitivity. They are sometimes crossed with sensitization to peanuts and tree nuts. There is evidence that dermal exposure is more effective to food than the intragastric, intranasal, or sublingual routes in human and animal trials. This suggests that skin contact with food may be a potent food sensitizer. Conversely, antigen uptake through intact skin has also been shown to regulate antigen-specific responses negatively. These studies demonstrated that the epicutaneous sensitization with food antigen results in an expansion of Treg with PD-1/PD-L1 exposure and the production of tolerance to the product when there was prior induction in MLN and mast cells or intestinal DCs, and when not, the presence of IgE-mediated anaphylaxis following the oral challenge with prior induction in the skin [121].

The relationship of the PD-1/PD-L1 inhibitory checkpoint in anaphylaxis has not been well studied. In 2017, Bonamichi-Santos R. studied the crucial role in activating the Th2 immune response profile and expression in murine models, suggesting that PD-L1 plays an important role. The animal model of anaphylaxis used the response to induction with OVA. They used blockade of PD-L1 by an anti-PD-L1 monoclonal antibody during sensitization, which significantly decreased specific IgE. PD-L1 ligands are strongly expressed in murine CM, but few studies have investigated their role in allergic reactions, especially anaphylactic reactions. This is the first study to evaluate the action of this protein in an allergy model for anaphylaxis and focuses on the immediate phase of the Gell and Coombs type I hypersensitivity reaction [103]. Other studies have attempted to elucidate the role of PD-1 pathways, induced by binding to both PD-L1 and PD-L2, in murine models of allergic respiratory disease. In a review article, based mainly on respiratory allergy models, the authors concluded that the PD-1/PD-L1 interaction appears to induce a Th2 response, with an increase in IL-4, whereas the PD-1/PD-L2 interaction induces a Th1 response with up-regulated IFN- up-regulation. Therefore, it was suggested that simultaneous expression of PD-L1 and PD-L2 ligands might neutralize these effects and not cause polarization [77]. Blockade of PD-1/PDL1 interaction with anti-PD-L1 results in weaker polarization toward Th2, with lower IgE and IgG1 [122]. However, these results were not convincing because the serum level of specific IgE influences MCs sensitization, explaining the observation of fewer degranulated MCs and lower allergic reaction in the group treated with anti-PD-L1, a failure in the use of monoclonal antibody with repeated injections.

## 5. Conclusions

There is increasing evidence that the PD-1 and its ligands axis play an important role in allergic disease. The binding of PD-1 by its ligands leads to a cascade of intracellular signaling that results in a range of immunoregulatory functions in T cell activation, tolerance, and immune-mediated disease. Regarding therapeutic approaches, blocking antibodies against PD-1 or its ligands has revolutionized cancer immunotherapy. However, to date, there are no controlled trials to determine therapeutic interventions of the PD-1/PDL axis in allergic disease. Promising evidence has been obtained from animal studies showing that using a PD-1 agonist significantly reduces airway hyperreactivity and lung inflammation, as demonstrated by Helow et al. On the other hand, PD-1 ligands seem to differentially regulate the immune response: PD-L2 exhibits a suppressive effect by downregulating IL-4 while up-regulating IFN-γ, resulting in a diminished AHR. In contrast, PD-L1 has the opposite effect as it up-regulates IL-4 and downregulates IFN-γ favoring a Th2 phenotype, which increases AHR [67]. These observations suggest that modulating PD-1 ligand-mediated pathways, blocking PD-L1, or activating PD-L2, might be a promising target in allergic asthma. There is also evidence of the involvement of the PD-1 and its ligand axis in other allergic diseases. For example, the demonstration that PD-L1 deficiency in a murine model of dermatitis leads to severe changes in the thickness of the ears and inflammation gives unequivocal proof for the involvement of PD-L1 in the pathogenesis of this disease. Future research will elucidate the regulatory pathways that control the PD-1 and its ligands expression to develop new drugs to treat allergic diseases.

## Figures and Tables

**Figure 1 ijms-22-11898-f001:**
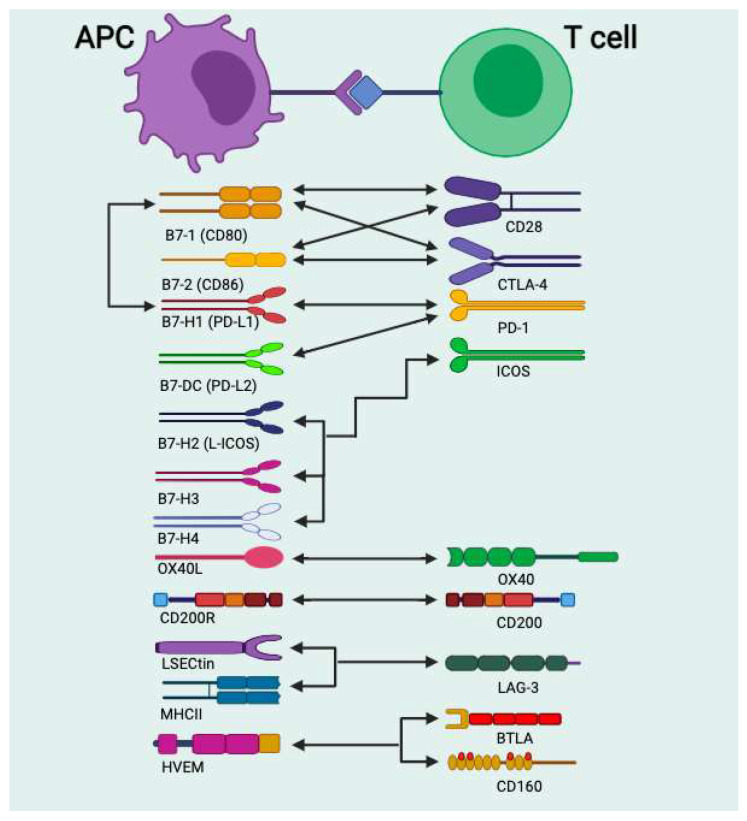
Stimulatory and inhibitory receptors in APCs and antigen-specific T cells. The receptor molecules and their corresponding ligands are shown. CD28:CD80/CD86 and CTLA-4:CD80/CD86 are co-stimulatory molecules. PD-1:PD-L1/PD-L1 CTLA-4, BTLA/HUVEM, and ICOS/ligands are coinhibitory molecules. Arrows show the interaction between receptors and ligands.

**Figure 2 ijms-22-11898-f002:**
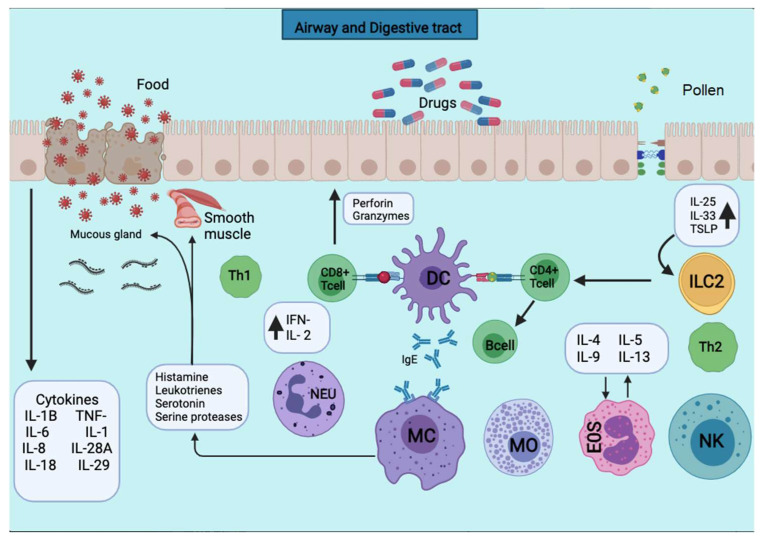
APCs present the antigens to the CD4+ T cells subpopulation for activation. APCs take up the first sensitizing contact with the antigen, bind it to MHC class II molecules, and expose it to the membrane. APCs take up the first sensitizing contact with the antigen, which processes and binds it to MHC class II molecules and exposes it to the membrane. Several cytokines such as TNF-a, IL-1, IL-4, IL-5, IL-6, IL-13, MIP-1a, and various colony-stimulating factors such as GM-CSF and IL-3 are released. The triggering of the allergic reaction is stimulated by the production of IgE by B lymphocytes. The production of IgE stimulates the triggering of the allergic reaction by B lymphocytes. The IgE will bind to mast cells and basophils for degranulation and the release of several mediators.

**Figure 3 ijms-22-11898-f003:**
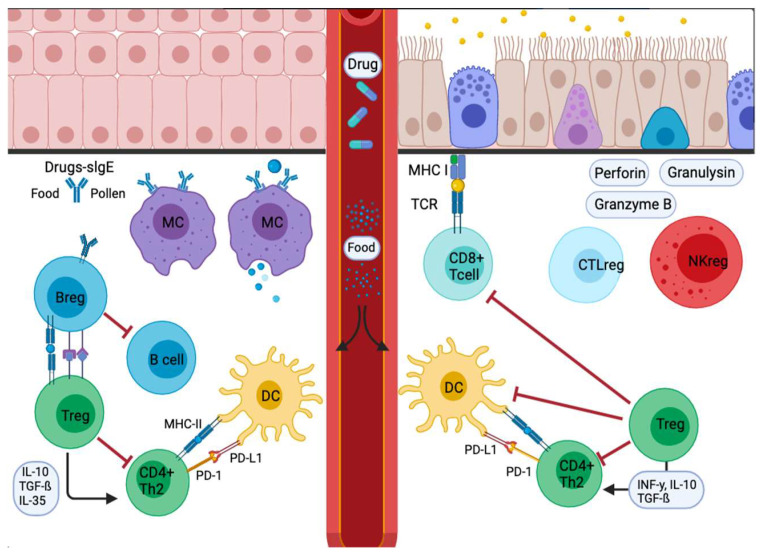
Signal inflammatory cascade allergy response in mucosas. Incoming allergens are taken up and processed by APCs or antibodies. Allergens cross the mucosa, then move to the lung pleura or intestine. In the sensitization stage, APCs present the antigen and give an immediate response. Others migrate to the draining lymph nodes, where the processed allergen is presented to naïve T and B cells, responses characterized and influenced by the cytokines secreted and the presence or absence of cell-bound co-stimulatory molecules. Decreased stimulus and/or the presence of IL-10, IL-35, or TGF-β triggers Treg cell transformation. In addition, APCs lead to the production of cytokines that regulate the change and expression of coinhibitory receptors such as PD1/PD-L1.

## Data Availability

Not applicable.

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
