# Peer review of "New Insights into the Role of PD-1 and Its Ligands in Allergic Disease"

_ijms, 2021, doi:10.3390/ijms222111898_

Round 1

Reviewer 1 Report

The review by Morales et al is focused on PD-1 and its ligands in the context of allergic diseases. The review is very interesting, because it deals with an under-investigated subject in the PD-1/PD-L1 field. The manuscript is of sufficient quality to be published after a major revision as described below:

It is evident that the authors are expert in allergic diseases, but they are not experts in PD-1 and PD-L1 biology. Most of the papers that are cited are outdated, considering the vast amount of literature on the field. They have left out many papers on PD-1 and PD-L1 biology that are key in the subject. Many of the concepts for PD-1 and PD-L1 described throughout the text are outdated or incomplete. Therefore, I recommend publication after a major revision. The key issues to correct are shown below, point by point:

  1. In the abstract the authors say: “

 However, the role of PD-1 and its ligands in the inflammatory process of allergic diseases is poorly understood. The main role of PD-1 and its ligands is to balance the immune response. These stimulatory molecules can function as a brake or a kick-start to regulate the adaptive immune response.”

PD-1 and it ligands are not truly considered stimulatory molecules. I would change this term to immune checkpoint molecules in the immunological synapse. This is the most accepted term.

  1. In introduction page 2, the authors say: “In addition, type-2-driven inflammation is characterized by high cellular plasticity that enables the cells to adapt to a specific inflammatory milieu.”

Here the authors need to explain with more detail what they mean with high cellular plasticity. Are they talking about T cells, myeloid cells? Are they referring to phenotypic plasticity? Proliferation capacities? In what context?

  1. In introduction page 2, the authors say:” Within the type 2 immune response complex endotype, several sub-endotypes might exist such as the IL-5-high, IL-13-high, or IgE-high endotype, and their predominance differs between allergic diseases”

The term “endotype” is not commonly used by most immunologists. At least this reviewer is not aware of it. The authors should define with some detail this term for the non-specialist reader.

  1. In page 2 line 62, the authors say “CD4+ T cells require two complementary signals to become fully activated. “

This is inaccurate. The consensus is that T cells require at least three types of signals to get fully activated unless they are memory T cells. The TCR signal, co-stimulation/inhibition signals and cytokine priming. Cytokine priming is particularly important because it determines CD4 T cell polarization. The authors need to include cytokine priming in their review.

  1. In page 2, line 80 the authors say: The CD3 core is expressed at the stable conjugation between”.

The correct term instead of CD3 core is the T-cell receptor (TCR) CD3 complex…

  1. In page 2, line 88: the authors say” Research to better characterize the specific role of these molecules in allergy and asthma has not been sufficient.”

Instead of saying “has not been sufficient”, I would say “it has not been the main focus of research in PD-1/PD-L1 biology”.

  1. In page 3, line 96 the authors say” and binding to antibodies attached to MCs, basophils, or eosinophils.”. I guess that MC stands for Mastocytes. Make sure all the acronyms are properly defined the first time they are used.
  2. In page 4, line 117 the authors claim:” Litle research on PD-1 and allergies exists in the literature, probably due to the recent discovery of the molecules described by Honjo and coworkers in 1992 [21].”.

I disagree. 30 years is not considered “recent”.  I would just say that the main focus of PD-1 and PD-L1 research has been classically cancer, and allergies has been under-investigated. In addition, PD-L1 should also be mentioned. It is as important as PD-1, and it was described by Lieping Chen. He did not get the Noble Prize, but many in the field believe he deserved it. Therefore, his work should be mentioned. The discovery of PD-L1 was made in 1999 and referenced here (PMID: 10581077).

  1. In page 4, line 122 the authors say: “PD-1 has been related as an inhibitor of immune responses, because PD-1–deficient mice developed autoimmune phenotypes [28].”.

The authors imply in this sentence that the role of PD-1 in regulating immune responses is still under question. This is clearly updated, due to the vast majority of literature, and the successful application of immune checkpoint inhibitors in routine clinical practice. I would rather define PD-1 as a modulator of immune responses, according to the vast majority of literature. Indeed, it plays a significant role in antigen presentation by DCs to T cells by regulating ligand-induced TCR down-modulation. This process does not lead to T cell inactivation, but ensures proper T cell activation. This is referenced in (PMID: 21739608).

  1. In page 4, line 131 the authors claim: “Under normal physiological conditions, PD-L1 mRNA was found expressed broadly in various tissues, but PD-L1 protein is only found on a few such as tonsil, a small fraction of macrophage-like cells, and various human cancer cells.”

It is highly inaccurate and very outdated. In physiological conditions, PD-L1 protein is highly expressed on the cell surface by the vast majority of systemic myeloid cells, including dendritic cells, and granulocytes. Its expression is upregulated after activation, but PD-L1 is high in these cells in peripheral blood. This is referenced in the paper above (PMID: 21739608), in this paper (PMID: 32973173), and in this clinical paper (PMID: 30986912). These are only just a few key examples from many. Indeed, the need for a more frequent administration of anti-PD-L1 atezolizumab antibody is because it is very quickly cleared by PDL1 high myeloid cells in circulation (PMID: 33849551).

  1. In page 4, line 135, the authors claim: “PD-L1 preferentially stimulates the production of IL-10 by T-cells. It may be involved in the negative regulation of T cell-mediated immune responses. [30]. “

No, PD-L1 does not preferentially stimulate the production of only IL-10 by T-cells. This is outdated and simplistic.  And its involvement as an inhibitor molecule for T cells is out of the question. The authors comment on this as if it was an unlikely possibility (“it may be involved in the negative regulation….”. PD-L1 is involved in fine-tuning immune responses, TCR downmodulation during physiological antigen presentation (PMID: 21739608), survival, glucose metabolism, proliferation of cells, including cancer cells, and T cells and migration of dendritic cells to the skin, through signalling by non-conventional domains. Its functions are so many and important, that I would encourage the authors to read a review on it so they can correct their statement (PMID: 30275987), and cite key papers on PD-L1 reverse signaling in cancer cells, dendritic cells and T cells (PMID: 32152508; PMID: 28834746; PMID: 33053342)

  1. In page 4, line 141 the authors claim “Recently was described that PD-1 ligands are also expressed in cancer cells [32] and 141 APCs from cancer tissues [33,34].”

Please revise this statement according to all the comments from above. This statement is outdated.

  1. In page 4, line 146, the authors claim: “Both in vitro and in vivo studies have proved that the expression of PD-L1 protein in APCs, regulatory T lymphocytes, and cancer cells strongly relied on the existence of IFN-.”

This statement is very incomplete. PD-L1 is expressed through two main mechanisms: By pro-inflammatory cytokines including type I and type II IFNs, TNF-alpha, IL-6, IL1beta, and the second mechanism driven by oncogenes. For references see (PMID: 28494868, PMID: 28834746, PMID: 28494868).

  1. In page 4, line 149 the authors say: “It is proposed that PD-L1 is up-regulated in response to inflammation and suppresses excessive immune responses, which may cause unnecessary tissue injury. Tumor cells that arise from normal cells adopt this mechanism to evade tumor immunity [38]”

This is incomplete. PD-L1 is also upregulated to inhibit apoptosis in many cells and regulate glucose metabolism. Please modify according to all the literature and information provided by this Reviewer.

  1. In page 7, line 256 the authors claim: “The binding of PD-L1 to PD-1 drives T cells to apoptosis or into regulatory phenotype [66]. PD-L1 has been characterized as a negative regulator of CD4+T cells.”

This is outdated. PD-1 is not an inducer of apoptosis, although it was firstly described as such. It is true that PD-1 constitutive signalling can lead to apoptosis in vitro but this is not its main function. Most of the times it regulates physiological T cell activation. But if PD-L1/PD-1 becomes chronic it leads to CD4 and CD8 T cell anergy and exhaustion (see papers PMID: 23610399 and  PMID: 31273938). It depends on the context of stimulation. In physiological situations, it leads to transient TCR down-modulation necessary for the exponential phase of T cell clonal expansion (PMID: 16724117 and PMID: 21739608). PD-L1 is not a negative regulator of only CD4+ T cells. It is also a negative regulator of CD8 T cells, which upregulate PD-1 strongly after antigen presentation (PMID: 29654146 and PMID: 21739608).

Author Response

Dear editors and reviewers:

Thank you very much for the evaluation of our manuscript and your constructive feedback. Your comments have been very helpful to improve our manuscript.

Following, we will give point-by-point answers to your comments:

Reviewer 1

1.- In the abstract the authors say: “However, the role of PD-1 and its ligands in the inflammatory process of allergic diseases is poorly understood. The main role of PD-1 and its ligands is to balance the immune response. These stimulatory molecules can function as a brake or a kick-start to regulate the adaptive immune response.”

PD-1 and it ligands are not truly considered stimulatory molecules. I would change this term to immune checkpoint molecules in the immunological synapse. This is the most accepted term.

LMT Response:

The term “stimulatory moleculesis changed to the suggested correct concept “immune checkpoint molecules” in the abstract and the rest of the revised manuscript.

  1. In introduction page 2, the authors say: “In addition, type-2-driven inflammation is characterized by high cellular plasticity that enables the cells to adapt to a specific inflammatory milieu.”

Here the authors need to explain with more detail what they mean with high cellular plasticity. Are they talking about T cells, myeloid cells? Are they referring to phenotypic plasticity? Proliferation capacities? In what context?

LMT Response:

The high cellular plasticity in the context of type-2-driven inflammation refers to the capacity of T-helper cell subpopulations to respond depending on environmental signals differentiating into other effector cell types. For example, in chronic asthma, Th1 and Th2 cytokines contribute to airway wall remodeling. Another group of cells that respond to environmental signals are group 2 innate lymphoid cells that exhibit a dynamic phenotype in allergic airway inflammation [10.1007/s12016-018-8712-1]. Based on the comments of the reviewer, this part was explained in the revised manuscript.

  1. In introduction page 2, the authors say:” Within the type 2 immune response complex endotype, several sub-endotypes might exist such as the IL-5-high, IL-13-high, or IgE-high endotype, and their predominance differs between allergic diseases”

The term “endotype” is not commonly used by most immunologists. At least this reviewer is not aware of it. The authors should define with some detail this term for the non-specialist reader.

LMT Response:

Allergic diseases such as asthma are heterogeneous with multiple conditions. The molecular phenotypes defined based on the predominant inflammatory cell are endotypes [10.1172/JCI124611]. In this context, the type 2 immune response can be categorized into three different subgroups, IL-5-high, IL-13-high, or IgE-high. Based on the reviewer recommendation we define the term in revised manuscript.

  1. In page 2 line 62, the authors say “CD4+ T cells require two complementary signals to become fully activated. “

This is inaccurate. The consensus is that T cells require at least three types of signals to get fully activated unless they are memory T cells. The TCR signal, co-stimulation/inhibition signals and cytokine priming. Cytokine priming is particularly important because it determines CD4 T cell polarization. The authors need to include cytokine priming in their review.

LMT Response:

Many thanks for the observation. We have now written that CD4+ T cells require three complementary signals to become fully activated. The TCR signal, co-stimulation/inhibition signals, and cytokine priming. Cytokines control differentiation into different types of effector cells deliver signal.

  1. In page 2, line 80 the authors say: The CD3 core is expressed at the stable conjugation between”.

The correct term instead of CD3 core is the T-cell receptor (TCR) CD3 complex…

LMT Response:

The revised manuscript corrected the sentence to “The T-cell receptor (TCR) CD3 complex is expressed at the stable conjugation between a T cell and APCs.” Thanks.

  1. In page 2, line 88: the authors say” Research to better characterize the specific role of these molecules in allergy and asthma has not been sufficient.”

Instead of saying “has not been sufficient”, I would say “it has not been the main focus of research in PD-1/PD-L1 biology”.

LMT Response:

The sentence was changed according to the reviewer´s suggestion: “Research to better characterize the specific role of these molecules in allergy and asthma has not been the main focus of research in PD-1 and its ligands biology.”

  1. In page 3, line 96 the authors say” and binding to antibodies attached to MCs, basophils, or eosinophils.”. I guess that MC stands for Mastocytes. Make sure all the acronyms are properly defined the first time they are used.

LMT Response:

The acronyms were defined the first time (line 97) correctly, MCs is for mastocytes in the revised manuscript.

  1. In page 4, line 117 the authors claim:” Little research on PD-1 and allergies exists in the literature, probably due to the recent discovery of the molecules described by Honjo and coworkers in 1992 [21].”.

I disagree. 30 years is not considered “recent”.  I would just say that the main focus of PD-1 and PD-L1 research has been classically cancer, and allergies has been under-investigated. In addition, PD-L1 should also be mentioned. It is as important as PD-1, and it was described by Lieping Chen. He did not get the Noble Prize, but many in the field believe he deserved it. Therefore, his work should be mentioned. The discovery of PD-L1 was made in 1999 and referenced here (PMID: 10581077).

LMT Response:

The text was re-written following the reviewer's suggestion. “PD-1 and its ligands research have been done principally in cancer, and allergies have been under-investigated. PD-1 was described by Honjo and coworkers in 1992, a few years later, in 1999, its ligand PD-L1 was described by Lieping Chen [10.1038/70932].” The DOI reference is now in the correct format in the final version of the manuscript.

  1. In page 4, line 122 the authors say: “PD-1 has been related as an inhibitor of immune responses, because PD-1–deficient mice developed autoimmune phenotypes [28].”.

The authors imply in this sentence that the role of PD-1 in regulating immune responses is still under question. This is clearly updated, due to the vast majority of literature, and the successful application of immune checkpoint inhibitors in routine clinical practice. I would rather define PD-1 as a modulator of immune responses, according to the vast majority of literature. Indeed, it plays a significant role in antigen presentation by DCs to T cells by regulating ligand-induced TCR down-modulation. This process does not lead to T cell inactivation, but ensures proper T cell activation. This is referenced in (PMID: 21739608).

LMT Response:

The focus on the role of PD-1 in regulating immune responses was changed to the modulator of immune responses. “PD-1 has been related as a modulator of immune responses, because PD-1–deficient mice developed autoimmune phenotypes [28]; also play a significant role in antigen presentation by DCs to CD8+ T cells by regulating ligand-induced TCR down-modulation [10.1002/emmm.201100165].”

  1. In page 4, line 131 the authors claim: “Under normal physiological conditions, PD-L1 mRNA was found expressed broadly in various tissues, but PD-L1 protein is only found on a few such as tonsil, a small fraction of macrophage-like cells, and various human cancer cells.”

It is highly inaccurate and very outdated. In physiological conditions, PD-L1 protein is highly expressed on the cell surface by the vast majority of systemic myeloid cells, including dendritic cells, and granulocytes. Its expression is upregulated after activation, but PD-L1 is high in these cells in peripheral blood. This is referenced in the paper above (PMID: 21739608), in this paper (PMID: 32973173), and in this clinical paper (PMID: 30986912). These are only just a few key examples from many. Indeed, the need for a more frequent administration of anti-PD-L1 atezolizumab antibody is because it is very quickly cleared by PDL1 high myeloid cells in circulation (PMID: 33849551).

LMT Response:

The statement on PD-L1 expression was changed: now reads as follows “PD-L1 protein is constitutively expressed on both hematopoietic and non-hematopoietic cells like DCs, granulocytes, T cells, B cells, and tumor cells [10.1038/ni1443, 10.1038/nri3790, s41467-020-18570-x]. A frequent administration of anti-PD-L1, like atezolizumab, is required for cancer treatment because PD-L1 is high in circulating-myeloid cells [10.3390/ijms20071631].” The DOI reference will be in the correct format in the final version of the manuscript.        

  1. In page 4, line 135, the authors claim: “PD-L1 preferentially stimulates the production of IL-10 by T-cells. It may be involved in the negative regulation of T cell-mediated immune responses. [30]. “

LMT Response:

The following text was included “PD-1 is an early brake that fine-tunes T-cell activation during antigen presentation after TCR signal transduction. PD-L1/PD-1 co-stimulation in antigen-presenting DCs contributes to PD-L1-induced TCR down-modulation [10.1002/emmm.201100165]. Also, PD-L1 stimulates the production of IL-10 by T-cells [10.1038/s41392-018-0022-9].

  1. In page 4, line 141 the authors claim “Recently was described that PD-1 ligands are also expressed in cancer cells [32] and 141 APCs from cancer tissues [33,34].”

Please revise this statement according to all the comments from above. This statement is outdated.

LMT Response:

Both paragraphs were deleted.

  1. In page 4, line 146, the authors claim: “Both in vitro and in vivo studies have proved that the expression of PD-L1 protein in APCs, regulatory T lymphocytes, and cancer cells strongly relied on the existence of IFN-g.”

LMT Response:

The statement has now been expanded “Both in vitro and in vivo studies have proved that the expression of PD-L1 protein in APCs, regulatory T lymphocytes, and cancer cells strongly relied on the existence of pro-inflammatory cytokines: IL-1b, TNF-alpha, IL-6, as well as IL-2, IL-7, IL-15, and IL-21. Also, IFNs type I and type II, and the second mechanism is driven by oncogenes (cyclin D-CDK4 cascade) [10.1111/j.1600-065X.2010.00923.x, 10.1016/j.celrep.2017.04.031,  10.1016/j.celrep.2017.07.075, 10.1016/j.celrep.2017.04.031, 10.1016/j.immuni.2018.03.014]”.

  1. In page 4, line 149 the authors say: “It is proposed that PD-L1 is up-regulated in response to inflammation and suppresses excessive immune responses, which may cause unnecessary tissue injury. Tumor cells that arise from normal cells adopt this mechanism to evade tumor immunity [38]”.

This is incomplete. PD-L1 is also upregulated to inhibit apoptosis in many cells and regulate glucose metabolism. Please modify according to all the literature and information provided by this Reviewer.

LMT Response:

The text was modified as suggested by the referee “The role of PD-L1 is to maintain peripheral tolerance and contribute to antigen presentation to T cells by dendritic cells [10.1016/j.celrep.2017.07.075]. PD-L1 is up-regulated in response to inflammation and suppresses excessive immune responses, which may cause unnecessary tissue injury. Tumor cells that arise from normal cells adopt this mechanism to evade tumor immunity. T-lymphocyte suppression for PD-L1 in the tumor microenvironment. Likewise, PD-L1 up-regulated to inhibit apoptosis in many cells and regulate glucose metabolism [10.1038/s41420-021-00401-7]”.

  1. In page 7, line 256 the authors claim: “The binding of PD-L1 to PD-1 drives T cells to apoptosis or into regulatory phenotype [66]. PD-L1 has been characterized as a negative regulator of CD4+T cells.”

This is outdated. PD-1 is not an inducer of apoptosis, although it was firstly described as such. It is true that PD-1 constitutive signalling can lead to apoptosis in vitro but this is not its main function. Most of the times it regulates physiological T cell activation. But if PD-L1/PD-1 becomes chronic it leads to CD4 and CD8 T cell anergy and exhaustion (see papers PMID: 23610399 and PMID: 31273938). It depends on the context of stimulation. In physiological situations, it leads to transient TCR down-modulation necessary for the exponential phase of T cell clonal expansion (PMID: 16724117 and PMID: 21739608). PD-L1 is not a negative regulator of only CD4+ T cells. It is also a negative regulator of CD8 T cells, which upregulate PD-1 strongly after antigen presentation (PMID: 29654146 and PMID: 21739608).

LMT Response:

Changes were made as suggested by the referee “PD-1 constitutive signaling can lead to apoptosis in vitro, but its primary function it regulate physiological T cell activation. However, if PD-L1/PD-1 becomes chronic, it leads to CD4 and CD8 T cell anergy and exhaustion and tolerance [10.1073/pnas.1305394110,  10.15252/emmm.201910293]. Moreover, as mentioned above, it led to transient TCR down-modulation necessary for the exponential phase of T cell clonal expansion in physiological situations. It needed to slow down a Ca(2+) flux and activation needed to slow down a Ca(2+) flux and activation [10.1038/sj.emboj.7601146, 10.1002/emmm.201100165]. PD-L1 is a negative regulator of  CD4+ and CD8+ T cells, which upregulates PD-1 strongly after antigen presentation [ 10.1073/pnas.1718217115, 10.1002/emmm.201100165]”.

Reviewer 2 Report

The programmed cell death 1 (PD-1) and its ligands PD-L1 and PD-L2 play important roles in the regulation of T cell activation.  In the review, “New insights into the role of PD-1 and its ligands in allergic disease”, the authors summarized the latest findings on how PD-1 signaling may affect Th2 activation (a fundamental mechanism of allergic diseases) in asthma, rhinitis, food allergy and skin allergic disease. Most of the results, as the authors point out, are extracted from animal models, together with a few clinical observations, and the effects of gene deletion or PD-1, PD-L1 and PD-L2 antagonists on Th2 polarization, IL4 responses and Th1 cytokines responses were individually analyzed. These findings suggest that PD-1, PD-L1 and PD-L2 act differently as co-stimulatory and co-inhibitory factors for Th2 responses. The modulation of the PD-1-mediated pathway by blocking PD-L1 or activating PD-L2 might be a promising target for allergic treatment.

The contents of review are quite timely even though the clinical significance needs to be validated.  More than 100 references are included in this review, including papers published in the last ten years. However, I would prefer a more coherent summary, rather than a repetition of all the results discussed for each disease in current “5. Conclusions”.  I believe this will improve the readability of the summary and the authors will be able to set forth their conclusions in a way that readers will appreciate.

Overall, I support its publication, but I do recommend the changes noted immediately above.

Author Response

Dear editors and reviewers:

Thank you very much for the evaluation of our manuscript and your constructive feedback. Your comments have been very helpful to improve our manuscript.

Following, we will give point-by-point answers to your comments:

The programmed cell death 1 (PD-1) and its ligands PD-L1 and PD-L2 play important roles in the regulation of T cell activation.  In the review, “New insights into the role of PD-1 and its ligands in allergic disease”, the authors summarized the latest findings on how PD-1 signaling…

The contents of review are quite timely even though the clinical significance needs to be validated.  More than 100 references are included in this review, including papers published in the last ten years. However, I would prefer a more coherent summary, rather than a repetition of all the results discussed for each disease in current “5. Conclusions”.  I believe this will improve the readability of the summary and the authors will be able to set forth their conclusions in a way that readers will appreciate. Overall, I support its publication, but I do recommend the changes noted immediately above.

LMT response:

Many thanks foy your comments and supporting the publication of our manuscript. Following your suggestions, we improved the conclusion (it was fully re-written).

Reviewer 3 Report

Thank you for giving me this opportunity to review this article. This article is well-written and provide insightful knowledge toward PD-1 and its ligands in several allergic diseases.

This article could be improved if the authors add more discussion into the animal models; that is, not only summarized the findings but also try to list the findings of each iconic study. Another suggestion is to provide as many clinical evidence in the therapeutic potential of PD-1 and its ligands in allergic diseases as you can. 

Author Response

Dear editors and reviewers:

Thank you very much for the evaluation of our manuscript and your constructive feedback. Your comments have been very helpful to improve our manuscript.

Following, we will give point-by-point answers to your comments:

Reviewer 3

Thank you for giving me this opportunity to review this article. This article is well-written and provide insightful knowledge toward PD-1 and its ligands in several allergic diseases.

This article could be improved if the authors add more discussion into the animal models; that is, not only summarized the findings but also try to list the findings of each iconic study. Another suggestion is to provide as many clinical evidence in the therapeutic potential of PD-1 and its ligands in allergic diseases as you can.

LMT Response:

Thank you for your helpful comments. We have included additional information on the animal models and provided more clinical evidence on the therapeutic potential of PD-1. Particular emphasis was placed on the conclusion section on this issue.

Round 2

Reviewer 1 Report

After the corrections, the paper should be accepted in the present form

Author Response

We appreciate your comments and contribution to our performance.